# Drivers of disengagement from care during the first six months on antiretroviral therapy for HIV in South Africa: A mixed-methods study

Mhairi Maskew[1]*, Nyasha Mutanda[1], Nancy Scott[2], Allison Morgan[2], Mariet Benade[1,2,3], Vinolia Ntjikelane[1], Linda Sande[1], Lufuno Malala[4], Musa Manganye[4], Sydney Rosen[1,2]

**1** Health Economics and Epidemiology Research Office, Faculty of Health Sciences, University of the Witwatersrand, Johannesburg, South Africa, **2** Department of Global Health, Boston University School of Public Health, Boston, Massachussetts, United States of America, **3** Department of Global Health, Amsterdam Institute for Global Health and Development, Amsterdam UMC, University of Amsterdam, Amsterdam, the Netherlands, **4** HIV/AIDS Treatment, Care and Support, South African National Department of Health, Pretoria, South Africa

\* mmaskew@heroza.org

## Abstract

For clients on HIV treatment in sub-Saharan Africa, early disengagement from care is a critical obstacle to achieving UNAIDS's second 95 target. While South Africa's Differentiated Service Delivery Guideline on Fast Track Initiation and Counseling (FTIC) defines normative procedures, the impact of implementation and drivers of disengagement remain unclear. The PREFER mixed-methods study enrolled a prospective cohort of adult clients initiating ART, returning to care after disengagement, or on ART for ≤6 months at 18 public healthcare facilities in South Africa. A survey collected demographic, clinical, and preference data. Participants were followed for up to 7 months using routinely collected medical records to estimate continuity of care (attendance at all scheduled visits within 28 days). Focus group discussions (FGDs) were conducted 12 months post-enrollment among participants who had expressed concerns about retention. A content analysis was conducted; emergent themes were situated within the Socio-Ecological Model framework. During the study period 7/9/2022–30/6/2023, PREFER-SA enrolled 1,049 participants (72% female, median age 33 years, 24% with CD4 < 200 cells/mm³); 122 also joined FGDs. By 6 months on ART, 23% were not continuously in care. New ART initiators were more likely to experience treatment interruption than those already retained for ≥1 visit. Disengagement was more likely among men, younger clients (18–24 years), those reporting food insecurity, and those not initiated on dolutegravir regimens. No differences were observed by relationship status, CD4 count or preferences for HIV care. FGDs revealed barriers to retention across all levels of the SEM, especially related to facility experience. Among adults initiating or re-initiating ART in South Africa, the highest risk of disengagement occurs immediately after initiation. Several potentially

**Data availability statement:** Data generated by the study will be made publicly available in the Open BU repository (https://open.bu.edu/) after the PREFER study protocol has been closed (anticipated closure December 2026). Until then, data will remain under the supervision of the Boston University Medical Campus IRB and the University of the Witwatersrand Human Research Ethics Committee (HREC). Requests can be sent to the BUMC IRB at medirb@bu.edu. Data extracted from routine medical records are owned by the study sites and the South African National Department of Health and cannot be made publicly available by the authors.

**Funding:** Funding for the study was provided by the Bill & Melinda Gates Foundation through INV-031690 to Boston University to MM, NM, NS, AM, MB, VN, LS and SR. The funder had no role in study design, data collection and analysis, decision to publish, or preparation of the manuscript.

**Competing interests:** I have read the journal's policy and the authors of this manuscript have the following competing interests. Drs. Manganye and Malala are employees of the government agency that has supervisory authority over the study sites.

modifiable individual and social factors were associated with early disengagement. Strengthening implementation of South Africa's Service Delivery Guidelines and improving facility experiences may support better retention in early treatment.

## Introduction

The successful implementation of universal access to antiretroviral therapy (ART) for HIV treatment has achieved global targets for reaching clients living with HIV in many African countries with high burdens of HIV. Despite these successes, optimal outcomes for HIV treatment have not yet been achieved, due to losses from care after ART initiation. The first six months after ART initiation, dubbed the "early retention" period [1], remain the period of highest risk for interruption or disengagement [2]. In a recent study in South Africa that relied on routinely collected electronic medical record (EMR) data, 16% of ART initiators had disengaged by 6 months. Another 14% had experienced an interruption of more than 28 days during that period, before returning to care, and an interruption in the early treatment period was a strong predictor of disengagement between 7 and 12 months, with a relative risk of disengagement of 1.76 compared to those who had not experienced a treatment interruption [3].

In addition to the challenges associated with starting any new kind of chronic medical care, achieving retention in care during the early treatment period requires overcoming some specific barriers. Clients presenting for ART initiation fall generally into three categories: 1) Treatment-naïve clients presenting for treatment initiation with CD4 counts <200 cells/mm$^3$ [4,5] or in WHO stage III or IV requiring additional clinical care for advanced HIV disease (AHD) [6] (despite universal access to HIV treatment, this group continues to represent a quarter to a third of clients presenting for treatment initiation); 2) Treatment-naive clients presenting for initiation with higher CD4 counts (≥200 cells/mm$^3$) and in WHO stage I or II. Starting CD4 counts have climbed steadily since the introduction of universal treatment in 2016 for those who do not have AHD [7]. In the contemporary era these clients, who are the prototypical patient for whom guidelines were originally designed, likely comprise only a third or fewer of those presenting for treatment initiation; and 3) Clients who have prior ART experience and are re-initiating ART after previously disengaging from care. Recent studies indicate that half or more of those who present for "initiation" in South Africa have prior ART experience [8,9]. Individuals who know their status and have already started and stopped ART at least once (re-initiators) may face the same barriers to retention again, if they have not received intervention related to those barriers. Re-initiators often present with AHD as well [4–9].

Because of their high risk of disengagement, all three of the categories of early treatment clients are important and require more attention. In addition, neither of the major treatment cascade-related service delivery innovations of the past decade--same-day initiation of ART [10] and differentiated models of HIV care for clients established and stable on ART [11]--address the early treatment period. Research

on retention in HIV care, moreover, has generally focused on long-term attrition from care, regarding the first six months as a uniform period that marks the shortest interval before a measurable outcome (retention or viral suppression) can be assessed [2]. Research is needed to identify and understand potential drivers of disengagement during the early treatment period so that client-appropriate interventions can be developed. Here, employing a mixed-methods approach to characterize the early treatment period in South Africa, we identify associations between client characteristics and early treatment outcomes in the PREFER study and explore client perceptions of barriers to adherence in their first six months on treatment.

## Methods

### Ethics statement

The PREFER study was approved by the Boston University Institutional Review Board (South Africa H-42726, May 20, 2022) and by the University of the Witwatersrand Human Research Ethics Committee (South Africa M220440, August 23, 2022). The protocol for South Africa was approved by Provincial Health Research Committees through the National Health Research Database for each study district (August 1, 2022 for West Rand; September 1, 2022 for King Cetshwayo and August 28, 2022 for Ehlanzeni). The study is registered with ClinicalTrials.gov (NCT05454839).

### Study design

PREFER was an observational, prospective cohort study conducted in South Africa (7 September 2022 - 30 June 2023) that aimed to describe and understand the needs of clients both newly initiating and re-initiating ART in order to inform the design of DSD models for the early HIV treatment period and improve early treatment outcomes [12]. During the study period, clients presenting for ART initiation or re-initiation were enrolled during a routine HIV care clinic visit and administered a baseline survey with quantitative and open-ended questions. As the PREFER study aimed to understand client perceptions and experiences of HIV care throughout the first six months on ART, the study also enrolled participants who had initiated ART up to 6 months prior to the study enrollment (i.e., clients enrolled while attending any clinic visit after the initiation visit but within 6 months of initiation) to ensure that the experiences and challenges of clients who had been accessing HIV care service delivery would be observed. Follow up included passive medical record review and participation by a subsample of survey responders in focus group discussions (FGDs).

### Study sites and population

PREFER was conducted at 18 public-sector healthcare facilities across three provinces in South Africa (Gauteng, KwaZulu-Natal and Mpumalanga) that had large ART client volumes and utilized the national electronic medical record system for HIV (TIER.Net). Sites were selected that were geographically accessible and jointly provided diversity in setting (rural, urban), facility size, and nongovernmental support partners. The study sequentially enrolled clients that were ≥18 years old, initiating, re-initiating, or on ART for ≤6 months and were able to provide written informed consent. Consent included participation in the baseline survey, permission to access medical records, and agreement to be contacted for follow-up within 12 months if more information was sought. During screening, clients who were unwilling to take the time required to complete the survey on the day of consent, unable to communicate in one of the languages into which the survey was translated, or judged by clinic or study staff to be physically or emotionally unable to provide consent or participate in all study procedures were excluded.

During the period of PREFER enrollment, the 2019/2020 South African ART guidelines were in place. They recommended a standard first-line ARV regimen of tenofovir, emtricitabine and dolutegravir [13,14]. Standard of care for the first six months on ART South Africa included 5–7 facility visits for clinical consultation, laboratory tests, and medication dispensing, with the first routine viral load (VL) test conducted at 6 months post initiation. Eligibility for less intensive models

of care (sometimes known as differentiated models) required at least 6 months' experience on ART and documentation of a suppressed viral load, making early treatment clients ineligible for South Africa's repeat prescription strategies [13,14]. (We note that new guidelines were issued in 2023 which revised the schedule for the early treatment period and eligibility criteria for less intensive models of care [15]; these are discussed further below.)

## Data sources, recruitment, and enrollment

The PREFER survey instrument was an interviewer-administered structured questionnaire that collected self-reported data on a client's demographic characteristics and socio-economic status, HIV testing history, HIV treatment history, current HIV care and treatment experience, other healthcare, preferences for treatment delivery, expectations, and costs of seeking care [12]. Participants were then passively followed up through EMR data from TIER.Net and South Africa's National Health Laboratory Services database and from paper records and registers maintained at the study sites. At the study sites, clinic staff informed potentially eligible clients that they may be eligible to participate in a research study when the patient checked in at the reception desk. Clients were recruited consecutively as they arrived at each facility, based on availability of study interviewers.

Roughly one year after the baseline survey, a subset of those who consented to follow-up were invited to participate in a qualitative FGD to explore participant experience in the early treatment period. Approximately 15 FGDs, each with 8–12 participants, were conducted in selected PREFER study facilities chosen to capture both urban and rural populations. We aimed for a maximum of 180 participants, a sample size that was expected to reach saturation in interview themes, after which no new insights would reasonably be expected to emerge. FGD participants were purposively invited based on responses to the PREFER quantitative survey that indicated they anticipated experiencing, or had already experienced, challenges to adherence in the first six months. These included having missed a scheduled clinic visit by 2–3 days in the past, anticipating difficulty in taking daily medication or picking up medications, or reporting that the care that they received in the clinic was worse than they had expected. FGD guides were designed to elicit perspectives on the challenges faced by clients and factors that enabled adherence in the early treatment period [12]. Focus groups were conducted by two study research assistants, one of whom served as the moderator while the second study research assistant observed and took notes.

## Outcomes and statistical analysis

The primary outcome for this analysis was retention in HIV care by 6 months after ART initiation. We classified retention into three patterns of engagement derived from observed visit attendance at scheduled clinical or medication pickup visits during the first 6 months after initiation: (i) ***continuously in care*** if all scheduled visits during the first 6 months after initiation were observed within 28 days of scheduled visit date or the client record indicated a documented transfer to another care facility prior to the 6-month outcome date; (ii) ***not continuously in care-disengaged*** if no visits were observed in the EMR after ART initiation, or at least one additional visit post initiation was observed but the last scheduled visit was not attended and no further visits were observed in the EMR, or a known date of death was documented in the EMR; or (iii) ***not continuously engaged in care--cyclical*** if at least one visit post initiation was attended >28 days after the scheduled visit date (Table 1).

Our secondary outcome was HIV viral load (VL) suppression by 6 months on ART. To account for newly updated South African guidelines recommending VL testing as early as 3 months after ART initiation, which came into effect during our enrollment period [15], we included any VL test result observed in months 3–7 after initiation. We then classified each viral load result observed into one of three suppression categories: 1) VL suppressed (VL test result <50 copies/mL); 2) low grade viraemia (VL test result 50–1000 copies/mL; or 3) VL unsuppressed (VL test result >1000 copies/mL).

We first describe characteristics and crude outcomes of enrolled participants using frequencies and simple proportions for categorical variables and medians with interquartile ranges for continuous variables. We then report the crude analysis showing simple proportions with 95% confidence intervals of patients not continuously in care at 6 months on ART and VL

Global Public Health PLOS

**Table 1. Retention in care outcome definitions.**

| Pattern | Sub-pattern | Definitions |
|---|---|---|
| Continuously in care | In care | All scheduled visits during the first 6 months after initiation observed within 28 days of scheduled visit date |
| | Transferred | Documented transfer to another care facility prior to 6-month date |
| Not continuously in care--disengaged | Died | Documented date of death in EMR prior to 6-month date |
| | Immediately disengaged | No visit after the ART initiation visit observed in the EMR |
| | Disengaged from care during first 6 months on ART | ≥1 visit after ART initiation visit observed in the EMR; last scheduled visit in 6-month period not attended |
| Not continuously in care--cyclical | Treatment interruption followed by re-engagement in care by 6 months on ART | ≥1 visit after ART initiation visit attended >28 days late during the first 6 months on ART |

suppression results. Next, we present a simple unadjusted comparison of outcome groups (continuously or not continuously in care at 6 months) with respect to baseline predictors of outcomes. Potential predictors included demographic and clinical variables, preferences and experiences reported in the PREFER survey as well as geographic and facility-level factors (urban vs rural setting, facility type). Finally, we use a log-linear regression model to estimate crude risk ratios and their corresponding 95% confidence intervals and utilize the change-in-estimate approach to estimate an adjusted model.

## Sensitivity analysis

As the PREFER study enrolment strategy included participants who had initiated ART up to 6 months prior to the study enrollment, we note the potential for selection bias in two important ways for the estimation of retention outcomes outlined in this analysis.. First, clients who have remained in care after the initial treatment visit have, by definition, been retained in care through a critical period for disengagement and thus may be very different from those who are newly initiating or newly re-initiating treatment. Second, clients who have been in care for 4–5 months have very little time for the outcome of interest (care disengagement within 6 months) to occur. In order to address this source of potential bias, we conducted sensitivity analyses stratifying the overall primary analysis by client ART exposure at study enrolment (newly initiating, re-engaging after an interruption or on ART (<6 months) to examine whether outcomes of clients who are already on ART for <6 months yields consistent results to those reported.

## Qualitative analysis

For qualitative data analysis, we used a combination inductive-deductive approach to develop a codebook to align the FGD guide, the quantitative survey, and the published literature. Two coders coded an initial set of transcripts and compared coding to ensure consistency. Discrepancies were resolved together, and one coder completed coding the remaining transcripts.

We conducted a content analysis and situated the emergent themes within the Socio-Ecological Model (SEM) framework to understand barriers and facilitators within and between individual, interpersonal, organizational, and community levels [16]. The SEM positions individuals within layers of influence [17], such as individual (e.g., one's age, income, health history, etc.); interpersonal (e.g., family relationships and interactions, social support); community (e.g., social networks, community sentiments, belonging, built environment); and societal (e.g., policy, social norms, economic opportunity, etc.). Such influences interact with each other in positive and negative ways and together shape peoples' lived

experiences. The model has been used by researchers and stakeholders to understand contributors to poor health and identify opportunity for health delivery and promotion [18]. We adapted the McLeroy's SEM for our population to identify barriers at the individual, interpersonal, health facility, and community levels to understand how facility operations and community sentiments affect clients' care-seeking behaviors. Qualitative findings did not differ between urban and rural sites, and are therefore presented together. Emergent FGD themes are presented with accompanying illustrative quotes and triangulated with quantitative findings within the results and discussion. Direct quotes were edited lightly for grammar and clarity.

## Results

### Characteristics of enrolled study participants

During the study enrollment period (7 September 2022 - 30 June 2023), we screened a total of 1,115 potential participants and enrolled 1,098 (the analytic sample; Fig 1). All potentially eligible participants were requested to provide written informed consent to participate; clients who refused consent (n = 1) or had enrolled previously (n = 11) were excluded, as were those who had been on ART for >6 months at screening date. An additional 2 clients were enrolled but their survey data were not completed and they were excluded from further analysis. A total of 1,049 participant records were successfully linked to the EMR used to ascertain 6-month retention outcomes and were included as the final linked sample. We identified 481 clients eligible to participate in focus group discussions, of whom 155 were contacted; 122 were enrolled in 15 FGDs and 32 declined. Reasons for not participating included working and not available during the week (n = 10);

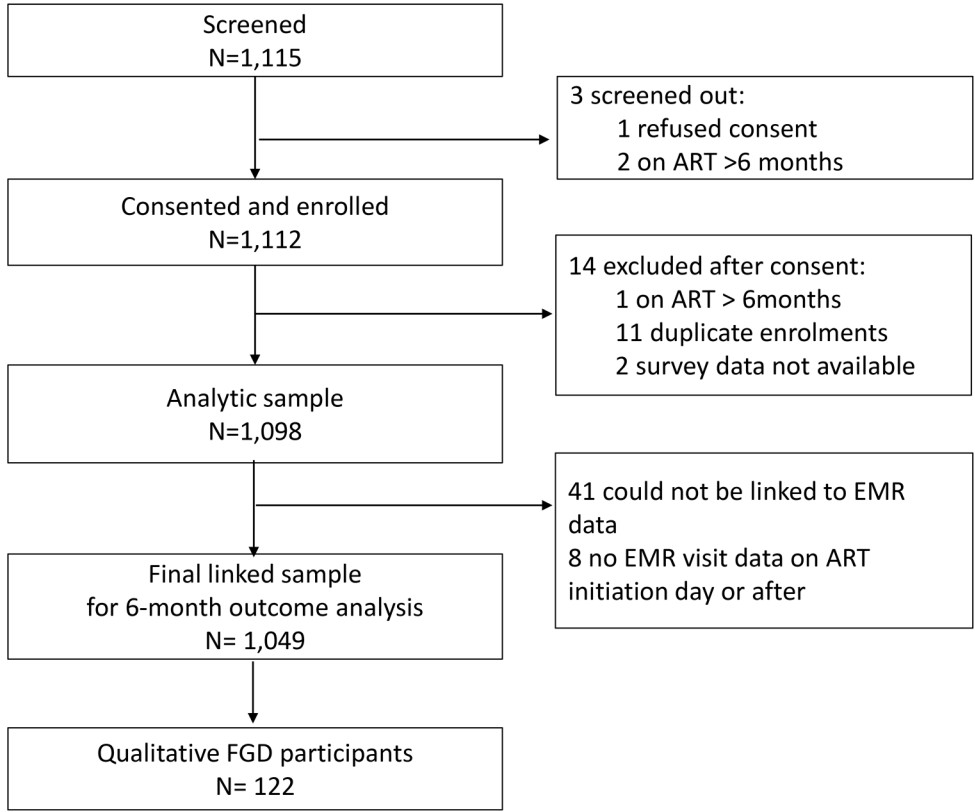

**Fig 1. PREFER study enrolment chart.**

relocation (n = 9); unable to confirm a date available (n = 5); not interested (n = 3); medical reasons (n = 2); and failure to arrive despite accepting (n = 3).

Table 2 presents the demographic and socioeconomic characteristics of 1,049 enrolled participants (final linked sample) included in the 6-month outcomes analysis. This final linked sample (n = 1,049) did not differ substantively from the analytic sample (n = 1,098) with respect to baseline covariates. A majority of participants were female (72%); the median age was 33 years (IQR 27–40 years). More than half of participants were unemployed and reported that access to money for healthcare would be difficult. Over 40% were initiating or reinitiating ART on the day of study enrolment, and a quarter had presented for ART initiation with a CD4 count <200cells/mm³. FGD participants were primarily female (84%) and had generally similar demographic characteristics to the full analytic sample, despite being deliberately selected because they had anticipated or experienced barriers to adherence.

### Retention and viral load suppression outcomes

At 6 months after ART initiation, 815 of the 1,049 participants (77%) were considered continuously in care, 92 (9%) had disengaged from care, and 142 (14%) had a cyclical pattern of engagement (Table 3). The proportion continuously in care included 45 individuals (4%) who had formally transferred to other facilities. Disengagement was comprised of immediate disengagers (no visits after initiation, 3%) and those who did not return after the first post-initiation visit (6%), along with a small number of deaths (0.5%). Cyclical engagement was common, as anticipated from previous research [3].

Out of the 1,049 participants, 705 (67%) had a VL test result available. Those without a VL test result available were similar to those with a test result in terms of age, gender, employment status (S1 Table) but were less likely to have secondary or tertiary education, to have/be living with a partner and able to obtain cash for healthcare. A third of participants (33%) did not have a VL test during the first 6 months on ART, due largely to early mortality, transfer, or disengagement from care. Cyclical engagers were also less likely than continuous engagers to have a VL test despite returning to care after a treatment interruption and were less likely to achieve viral suppression among those who had a viral load.

#### *Sensitivity analysis*

We further stratified the retention and viral load suppression outcomes by ART status at study enrolment (S2–S4 Tables). The proportion of clients continuously in care by 6 months on ART was lowest among newly initiating clients (70%) compared to re-engaging (80%) or those on treatment <6 months (81%). Among those participants who had a VL test, rates of VL suppression to <50 copies/mL were higher among those newly initiating (73%; 119/162) and those on treatment <6 months at study enrolment (71%; 311/441) compared to re-engaging clients (67%; 68/102).

### Predictors of retention outcomes

We found several potential predictors of retention at 6 months after initiation (Table 4). Women had a lower risk of disengagement during the first six months than their male counterparts (adjusted relative risk (ARR) 0.76; 95% CI 0.59-0.98). Those who reported experiencing food scarcity were more likely to have interrupted treatment compared to clients who reported consistent access to food (RR = 1.31; 95% CI 1.02-1.67). Clients who reported newly initiating ART had a higher risk (RR = 1.47; 95% CI 1.02-2.12) of not continuing in care than did those who had attended at least one post-initiation visit at the time of study enrolment. Though our estimates lacked precision, results also suggested that clients with CD4 counts <200 cells/mm3 at initiation were less likely to disengage from care at 6 months than those with higher CD4 counts at ART start (19% vs 23%; RR = 0.79; 95% CI: 0.58 - 1.07). Young adults aged 18–24 years also appeared more likely to not be continuously in care at 6 months on ART compared to older clients (27% vs 21%; RR = 1.24; 95% CI 0.93 - 1.65). Disengagement was higher among clients on an Efavirenz-containing regimen than for those on Dolutegravir (RR = 1.79; 95% CI 1.21-2.66). We did not observe important differences in retention by level of education, or relationship status or other self-reported preferences or experiences of HIV service delivery.

**Table 2. Demographic and socio-economic characteristics of enrolled participants.**

| Characteristic | Final linked sample (n = 1,049) | Qualitative FGD sample (n = 122) |
|---|---|---|
| Age (median, IQR) | 33 (27, 40) | 33 (28, 41) |
| Gender (n, % female) | 750 (71%) | 101 (83%) |
| Marital status | | |
| Live with primary partner | 334 (32%) | 34 (28%) |
| Do not live with a partner | 488 (47%) | 59 (48%) |
| Do not have a partner/spouse | 227 (22%) | 29 (24%) |
| Education level | | |
| Primary or less | 395 (38%) | 38 (31%) |
| Secondary | 501 (48%) | 66 (54%) |
| Post-secondary | 153 (15%) | 18 (15%) |
| Employment status | | |
| Formal employment | 228 (22%) | 15 (12%) |
| Informal employment | 206 (20%) | 24 (20%) |
| Unemployed | 538 (51%) | 73 (60%) |
| Student/trainee | 77 (7%) | 10 (8%) |
| Food scarcity | | |
| Never/seldom | 805 (77%) | 90 (74%) |
| Sometimes/often | 244 (23%) | 32 (26%) |
| Access to R100 ($6) for healthcare if needed | | |
| Easy | 460 (44%) | 45 (37%) |
| Difficult | 589 (56%) | 77 (63%) |
| District | | |
| District 1 | 349 (33%) | 40 (33%) |
| District 2 | 350 (33%) | 32 (26%) |
| District 3 | 350 (33%) | 50 (41%) |
| Urban/rural facility | | |
| Rural | 481 (46%) | 50 (41%) |
| Urban | 568 (54%) | 72 (59%) |
| Pregnant status at enrolment | | |
| Female not pregnant | 643 (61%) | – |
| Female pregnant | 107 (10%) | – |
| Male | 299 (29%) | – |
| Time on Art at study enrolment | | |
| 0 – 3 months | 811 (77%) | 96 (79%) |
| 4 – 6 months | 238 (23%) | 26 (21%) |
| ART status at study enrolment | | |
| Self-reported newly initiating | 296 (28%) | 35 (29%) |
| Self-reported re-engaging | 157 (15%) | 21 (17%) |
| On treatment | 596 (57%) | 66 (54%) |
| WHO staging at initiation | | |
| Stage 1 | 904 (86%) | 96 (79%) |
| Stage 2 | 82 (8%) | 11 (9%) |
| Stages 3 and 4 | 41 (4%) | 4 (3%) |
| Unknown | 22 (2%) | 11 (9%) |

*(Continued)*

**Table 2.** (Continued)

| Characteristic | Final linked sample (n = 1,049) | Qualitative FGD sample (n = 122) |
|---|---|---|
| Baseline CD4 count | | |
| CD4 count <200 cells | 255 (24%) | 83 (68%) |
| CD4 count >200 cells | 686 (65%) | 21 (17%) |
| Missing CD4 count | 108 (10%) | 18 (15%) |
| Baseline regimen | | |
| Tenofovir//Lamivudine/Dolutegravir | 946 (90%) | 104 (85%) |
| Tenofovir/Emtricitabine//Efavirenz | 84 (8%) | 10 (8%) |
| Other | 19 (2%) | 0 (0%) |
| Unknown | 0 (0%) | 8 (7%) |

**Table 3. Retention and VL suppression outcomes stratified by pattern of engagement (n = 1049).**

| Outcome | Outcome definition | Overall (n, %) | VL test not observed | VL < 50 copies/mL | VL 50–1000 copies/mL | VL > 1000 copies/mL |
|---|---|---|---|---|---|---|
| **Total N** | | 1049 | 344 (33%) | 498 (47%) | 149 (14%) | 58 (6%) |
| **Continuously in care** | In care at originating facility | 770 (73%) | 169 (22%) | 432 (56%) | 130 (17%) | 39 (5%) |
| | Transferred | 45 (4%) | 40 (89%) | 2 (4%) | 3 (7%) | 0 (0%) |
| **Disengaged from care** | Died | 4 (0.5%) | 4 (100%) | 0 (0%) | 0 (0%) | 0 (0%) |
| | Immediately disengaged | 30 (3%) | 30 (100%) | 0 (0%) | 0 (0%) | 0 (0%) |
| | Disengaged during first 6 months on ART | 58 (5.5%) | 57 (98%) | 0 (0%) | 1 (2%) | 0 (0%) |
| **Cyclical engagement** | Treatment interruption followed by re-engagement in care by 6 months after ART initiation | 142 (14%) | 44 (31%) | 64 (45%) | 15 (11%) | 19 (13%) |

## Predictors of viral load suppression by 6 months on ART

We also explored predictors of achieving viral load suppression to <50 copies/mL within the first 6 months on ART among those who had a VL test available (S5 Table). Overall, men were five percentage points less likely to have a suppressed VL at 6 months on ART compared to non-pregnant women (66% vs 71%; RR = 0.93; 95% CI 0.83-1.04) while pregnant women had the highest rates of VL suppression (79%; RR = 1.09; 95% CI 0.95-1.25). Younger clients aged 18–25 were 5 percentage points less likely to achieve viral load suppression compared to clients >25 years (66% vs 71%; RR = 0.87; 95% CI 0.75-1.00). Clients presenting with AHD (CD4 cell count <200mm$^3$) were also less likely to suppress viral load by 6 months on ART compared to those with CD4 > 200 mm$^3$ (59% vs 74%; RR 0.80; 95% CI 0.70-0.91). Those who expressed feeling that health workers were too busy to listen to client's problems were 12 percentage points less likely to have a VL < 50 copies/mL (60% vs 72%; RR = 0.86; 95% CI 0.71 - 1.05) while those initiating a dolutegravir based regimen were more likely to suppress VL by 6 months on ART (74% for DTG vs 56% for Efavirenz-based regimens (RR = 0.78; 95% CI 0.59 - 1.02) and 55% for other regimens (RR = 0.72; 95% CI 0.42 - 1.24), respectively).

## Qualitative findings

When FGD participants were asked qualitatively about barriers to adherence in the first six months, several key insights emerged at each social ecological level, though the majority pertained to interpersonal and health facility levels (Table 5).

At the individual level, key barriers were primarily financial, though they affected continuity of care through different

**Table 4. Predictors of disengagement during the first six months on ART treatment.**

| Characteristic | Continuously in care (n=815) | Not continuously in care (n=234) | Crude RR (95% CI) | Adjusted RR (95% CI)* |
|---|---|---|---|---|
| Age | | | | |
| ≥25 years | 683 (79%) | 185 (21%) | Ref | Ref |
| 18–24 years | 132 (73%) | 49 (27%) | 1.27 (0.97 - 1.67) | 1.24 (0.93 - 1.65) |
| Gender | | | | |
| Male | 224 (75%) | 75 (25%) | Ref | Ref |
| Female | 591 (79%) | 159 (21%) | 0.85 (0.67-1.07) | 0.76 (0.59 - 0.98) |
| Educational level | | | | |
| Secondary/ tertiary | 512 (78%) | 142 (22%) | Ref | Ref |
| Primary or less | 303 (77%) | 92 (23%) | 1.07 (0.85 - 1.35) | 1.09 (0.87 - 1.38) |
| Relationship status | | | | |
| Have a partner | 259 (78%) | 75 (22%) | Ref | Ref |
| No partner/not living with a partner | 556 (78%) | 159 (22%) | 0.99 (0.78 - 1.26) | 1.00 (0.79 - 1.29) |
| Food scarcity | | | | |
| Never/seldom | 639 (79%) | 166 (21%) | Ref | Ref |
| Sometimes/often | 176 (72%) | 68 (28%) | 1.35 (1.06 - 1.72) | 1.31 (1.02 - 1.67) |
| Can obtain ZAR100 cash for health care when required | | | | |
| Easy | 360 (78%) | 100 (22%) | Ref | Ref |
| Difficult | 455 (77%) | 134 (23%) | 1.05 (0.83 - 1.32) | 1.04 (0.83 - 1.31) |
| Urban/rural facility | | | | |
| Urban | 441 (78%) | 127 (22%) | Ref | Ref |
| Rural | 374 (78%) | 107 (22%) | 0.99 (0.79 - 1.25) | 1.02 (0.81 - 1.29) |
| Pregnant status at enrolment | | | | |
| Female not pregnant | 505 (79%) | 138 (21%) | Ref | Ref |
| Female pregnant | 86 (80%) | 21 (20%) | 0.91 (0.61 - 1.38) | 0.89 (0.59 - 1.35) |
| Male | 224 (75%) | 75 (25%) | 1.17 (0.91 - 1.49) | 1.30 (1.00 - 1.68) |
| ART status at study enrolment | | | | |
| On treatment<6 months | 482 (81%) | 114 (19%) | Ref | Ref |
| Newly initiating | 208 (70%) | 88 (30%) | 1.55 (1.22 – 1.98) | 1.54 (1.20 -1.97) |
| Re-engaging | 125 (80%) | 32 (20%) | 1.07 (0.75 – 1.51) | 1.05 (0.74 - 1.49) |
| WHO staging at initiation | | | | |
| Stage1 | 703 (78%) | 201 (22%) | Ref | Ref |
| Stage2 | 63 (77%) | 19 (23%) | 1.04 (0.69 - 1.57) | 1.07 (0.71 - 0.61) |
| Stage 3 or 4 | 32 (78%) | 9 (22%) | 0.99 (0.55 - 1.78) | 0.97 (0.54 - 1.73) |
| Unknown | 17 (77%) | 5 (23%) | 1.02 (0.47 - 2.23) | 1.04 (0.48 - 2.26) |
| Baseline CD4 count | | | | |
| ≥200 copies | 525 (77%) | 161 (23%) | Ref | Ref |
| <200 copies | 207 (81%) | 48 (19%) | 0.80 (0.60 1.07) | 0.79 (0.58 - 1.07) |
| Missing | 83 (77%) | 25 (23%) | 0.99 (0.68 - 1.43) | 1.05 (0.72 - 1.54) |
| Baseline regimen | | | | |
| Tenofovir/Lamivudine/Dolutegravir | 742 (78%) | 204 (22%) | Ref | Ref |
| Tenofovir/Emtricitabine/Efavirenz | 58 (69%) | 26 (31%) | 1.44 (1.02 - 2.02) | 1.79 (1.21 - 2.66) |
| Other | 15 (79%) | 4 (21%) | 0.98 (0.41 - 2.35) | 1.18 (0.48 - 2.89) |

*(Continued)*

**Table 4.** (Continued)

| Characteristic | Continuously in care (n = 815) | Not continuously in care (n = 234) | Crude RR (95% CI) | Adjusted RR (95% CI)* |
|---|---|---|---|---|
| *Preferences* | | | | |
| Do you wish that this clinic offered you more/the same/less HIV information? | | | | |
| Same or less | 418 (78%) | 121 (22%) | Ref | Ref |
| More | 397 (78%) | 113 (22%) | 0.99 (0.79 - 1.24) | 1.00 (0.80 - 1.25) |
| Do you wish that this clinic offered you more/ the same/less counselling? | | | | |
| Same or less | 420 (78%) | 116 (22%) | Ref | Ref |
| More | 395 (77%) | 118 (23%) | 1.06 (0.85 - 1.33) | 1.06 (0.85 - 1.32) |
| I prefer to attend clinic visits | | | | |
| Alone | 705 (78%) | 195 (22%) | Ref | Ref |
| With someone | 110 (74%) | 39 (26%) | 1.21 (0.90 - 1.63) | 1.12 (0.82 - 1.51) |
| I prefer to receive HIV treatment outside the clinic | | | | |
| No | 289 (78%) | 81 (22%) | Ref | Ref |
| Yes | 526 (77%) | 153 (23%) | 1.03 (0.81 - 1.31) | 1.05 (0.83 - 1.33) |
| *Facility experience* | | | | |
| Some staff do not treat patients with sufficient respect | | | | |
| Disagree | 611 (78%) | 171 (22%) | Ref | Ref |
| Agree | 151 (76%) | 49 (24%) | 1.12 (0.85 - 1.48) | 1.15 (0.87 - 1.51) |
| The queues to see a doctor or nurse are too long at this facility | | | | |
| Disagree | 211 (77%) | 63 (23%) | Ref | Ref |
| Agree | 538 (79%) | 146 (21%) | 0.93 (0.72 - 1.20) | 0.94 (0.73 - 1.21) |
| In this clinic, you're able to talk to the doctors or nurses in private | | | | |
| Disagree | 97 (76%) | 30 (24%) | Ref | Ref |
| Agree | 704 (78%) | 197 (22%) | 0.93 (0.66 - 1.30) | 0.92 (0.66 - 1.29) |
| The health workers are too busy to listen to my problems | | | | |
| Disagree | 728 (78%) | 202 (22%) | Ref | Ref |
| Agree | 72 (73%) | 26 (27%) | 1.22 (0.86 - 1.74) | 1.25 (0.89 - 1.76) |
| The doctors and nurses discussed the treatment fully with me | | | | |
| Disagree | 43 (75%) | 14 (25%) | Ref | Ref |
| Agree | 767 (78%) | 220 (22%) | 0.91 (0.57 - 1.45) | 0.90 (0.56 - 1.44) |
| I find it easy to tell the health workers when I have missed taking my tablets | | | | |
| Disagree | 146 (78%) | 42 (22%) | Ref | Ref |
| Agree | 573 (78%) | 159 (22%) | 0.97 (0.72 - 1.31) | 0.95 (0.71 - 1.28) |
| The facilities are dirty | | | | |
| Disagree | 704 (78%) | 195 (22%) | Ref | Ref |
| Agree | 84 (74%) | 30 (26%) | 1.21 (0.87 -1.69) | 1.26 (0.91 - 1.75) |

*Adjusted for age, gender, ART status at enrollment and baseline CD4 count. Note that gender was removed from the pregnancy status predictor analysis and baseline CD4 count was removed from the WHO stage predictor analysis due to potential collinearity.

**Table 5. Qualitative barriers to adherence in the early treatment period, fit to the social ecological model.**

| Level | Barrier | Quotations |
|---|---|---|
| Individual | Financial cost of travel | "I have trouble with transportation money. My date arrives and I don't have money for transport. I even go and borrow from a friend." |
| | Food scarcity | "The medication intake requires one to eat food. When you do not have income to buy food it really becomes a problem. I am discouraged to collect my medication from the clinic because I know that I will not take it since there is no food. So I might as well stay without the pills because of the lack of income" |
| Interpersonal | Stigma deters people from disclosing their status to friends and loved ones; feelings of ostracization within families and social circles. | "When I started taking treatment things changed at home. They had a discussion about me from day 1. I was given my own plate and spoon. I had my own things that I must use separately for everything I do in the house. So I am known as "that person on treatment" at home."<br>"Sometimes it happens that in a family you are the only one like this. What scares me is that I am the only one in the family with this problem. I can't open up and talk to a person that is HIV negative and tell them I am positive. It would have been easier if somebody else was dealing with the same problem in my family."<br>"It is hard just to tell even friends what is going on and that you are on medication. Sometimes you end up defaulting just to impress your family or the partner. You want to be seen as innocent to your partner." |
| Health facility | Negative interactions with clinic staff; feeling shamed | "Sometimes nurses come to the clinic with their own moods and attitudes and it becomes difficult to ask them about anything. It would be nice if you can walk in happy and leave happy. What happens sometimes is that you walk in well and leave feeling sick because the nurse who was attending to you has left you feeling stressed. You are even struggling to take instructions."<br>"But the problem started with the nurse who tested me. When she finished testing me she was like "you are busy with boys here." She did not even know... So next time when they say go to the social worker, no thank you, because already you have judged me. You have categorised me. Even if I was running around with boys, there was no need for you to say that." |
| | Inefficient filing system consumes staff time; results in multiple files per client and long wait times | "The problem is the missing files at the clinics. They will come and ask when you last attended the clinic when it is clearly stated on the appointment card. I am not sure how files go missing, but they do. You sit for 30 minutes whilst they are still looking for your file. People behind you will pass you and go in. You end up being attended late. There really needs to be another strategy when it comes to the filing system."<br>"My second and third visit were the same. My file was missing, [so they make a] new file and on subsequent visits the [missing] files will be all together and you just wonder where they came from now. I had just to accept to have a lot of files." |
| | Clients perceive the lack of privacy at the clinic; fear of disclosure | "What makes people anxious about going to the clinic is meeting people they know that work at the clinic. They worry about them spreading the information to other people they know. They feel like hiding so that they will not be seen by people that know them at the clinic. It would help if clinic staff took the issue of confidentiality seriously. That is the reason people do not like going to the clinic." |
| Community | Work and opportunity costs of care seeking | "My clinic date was on the 16th. My boss does not care if it is your clinic date. He wants you at work. So I wanted to leave work early but I couldn't. When I got of the taxi, it was already 16h00. I figured it was too late to go to the clinic. On the next day I woke up early to collect my treatment. When I got there, I was number 2 but could not get assistance because they were prioritising people that came after me with appointment cards. I got angry and asked them what was happening and they said that I will be seen last because I missed the date."<br>"Sometimes I will get the temporary job. I can't tell my boss that I have a temporary job. Even if I am given the day, when I come to the clinic at 8 o'clock, sometime I will get the treatment at past 2 or near 3. So, I would decide that it is better to go next week because [if I don't work] I will stay without having something to eat and no money in my pocket and no nothing. Sometimes I will prefer not to come because I have no money. Plus the time I am going to spend at the clinic." |

mechanisms. Participants reported the cost of travel to the facility as a barrier that directly hampered their ability to obtain care and collect medication. They also described general food scarcity which limited their willingness and ability to take the medication for fear of side effects, consistent with quantitative data in Table 2 showing over half of respondents in the analytic and FGD samples were unemployed at baseline and nearly a quarter of respondents reported 'sometimes/often' experiencing food scarcity which was associated with an increased likelihood of disengagement in the first six months.

At the interpersonal level, stigma and non-disclosure, two inextricably linked concepts, emerged as key barriers. Participants reported experiencing changes in how they were treated within their own homes and identified feelings of shame

related to fear and gossip within their social networks. As illustrated in the quotes presented below, these feelings of stigma and shame can result in behaviors related to disengagement including non-disclosure or defaulting on treatment.

Most of the barriers that emerged were at the facility level, wherein respondents reported negative interactions with health facility staff, inefficiencies in clinic processes, and a fear of disclosure because of a lack of privacy as demotivating factors that contributed to disengagement in the early treatment period. Respondents were frustrated with the processes of having to start a new file at every visit, which extended their wait time both while the provider searched for the old file and then started a new file. Respondents also explained that they experienced punitive repercussions for showing up a day or two late, including being shouted at or demoted to the back of the queue. Others described situations in which the felt they were being shamed or judged for having HIV.

Finally, at the community level, barriers pertained to challenges associated with missing work multiple times for clinic appointments, as well as other opportunity costs associated with care-seeking. Participants described challenges negotiating time off from work to get to the clinic. This was a challenge both for those with permanent jobs and for those with temporary positions. Participants also described holding jobs that required them to travel which prevented them from getting to the clinic.

## Discussion

Reaching the second of UNAIDS's "three 95s" [19] --initiating and retaining 95% of known PLWH on ART--has remained challenging for high HIV burden countries like South Africa. Addressing high rates of disengagement from care after treatment initiation is likely the only effective pathway to achieving current targets [20]. To do this, obstacles to establishing and remaining continuously in care during the period with the highest risk for disengagement—patients' first six months on ART—must be identified and solutions to overcome these challenges explored.

Results of the PREFER study reported here confirmed high rates of disengagement during the early treatment period; overall nearly a quarter of participants were not continuously in care by 6 months after ART initiation, similar to rates of disengagement reported in other studies in South Africa [21] Sensitivity analyses confirmed newly initiating clients to have higher rates of disengagement from care than those who had been on treatment for 1 or more months. A cyclical pattern of engagement characterized by an interruption in treatment of >28 days followed by re-engagement in care was more frequently observed during the early treatment period than was outright disengagement from care. Cyclical engagers in our cohort were most likely to have unsuppressed viral loads (>1000 copies/mL) at their six-month monitoring point, suggesting that they were indeed not accessing medication during the interruptions. This is consistent with other findings from South African cohorts suggesting poor treatment outcomes for returning engagers; one study found less than a third of clients maintain a suppressed VL one year after re-engaging in care [22] while another reported a four-fold increase in mortality risk among those who interrupt care during the first 6 months on ART [23].

Quantitative results of our study identified some key groups that may require prioritization for retention interventions during the early treatment period. Not surprisingly, men, youth (18–24 years), and those newly initiating ART were groups most at risk for not being continuously on ART by 6 months after treatment initiation. The first two populations are well-described demographic categories at risk of disengagement [24,25], while estimates for the latter group are likely impacted by survival bias –those in the study who had already returned for at least one clinic visit at the time of study enrollment had already been retained in care longer than the "immediate disengagers." The stratified results of the sensitivity analyses provide estimates of each outcome by time on ART to address this potential source of error.

We also found that those initiating a dolutegravir-based regimen at initiation were more likely to be continuously in care at 6 months than were those on efavirenz-based regimens. Dolutegravir-based regimens have been shown to be effective at rapidly suppressing virus and durable in terms of resistance development; clients also report fewer side effects on dolutegravir than other drug classes available in South Africa [26,27]. Also, not surprisingly, but less well documented in prior research, we found that clients in our study who either self-reported prior use of ART or were re-engaging in care

at study enrolment were more likely to be receiving efavirenz-based ART. In this context, treatment regimen may be a proxy for prior treatment experience, as previously disengaging from care may be independently associated both with an efavirenz regimen and an increased risk for a subsequent treatment interruption if the factors leading to the first episode of disengagement are not addressed.

While these findings are consistent with other work and begin to identify groups to prioritize, they offer few practical solutions. Although they are considered the highest risk groups, an absolute majority of men, young people, and previous disengagers do remain continuously in care (i.e., continuous retention exceeds 50%) and do not require additional intervention to achieve viral suppression. There is a need to get beyond these general characteristics if we are to identify important and modifiable drivers of disengagement and find feasible solutions. PREFER made some progress in this direction. We found that food scarcity and the use of an efavirenz-based regimen were both associated with a significant reduction in continuous engagement. While our estimates lacked precision, we also observed that participants with unfavorable views of their treatment experience--long waiting queues and negative interactions with healthcare workers, such as clients who report re-engaging in care being attended to after all other booked clients--were also less likely to remain continuously engaged. Finally, as expected, we observed high immediate disengagement from care (no further visits after initiation), indicating that for many clients, the initiation visit itself offers the only chance to influence later outcomes.

The qualitative findings offered further insights in this regard, while also reinforcing existing evidence about the patient-provider relationship [28–30]. Despite two decades of societal experience with HIV treatment since the rollout of ART in South Africa, study participants reported that stigma and social ostracization remain major impediments to treatment continuity. Clients reported this barrier both in their homes and at facilities, where negative language and shaming of clients living with HIV persists and privacy may be compromised. Another key facility-level barrier to visit adherence that clients reported was the inefficiencies created by poorly organized filing systems that result in loss of client information, duplicate files, and excessive waiting times at each facility visit.

While societal stigma is not easily addressed, many of the facility-level barriers identified in our study are modifiable within the healthcare system. A systematic review of interventions to improve retention [31] found positive benefits of community-based service delivery and adherence clubs, bearing in mind that the models reviewed were underpinned by supportive interventions that addressed psychosocial barriers to adherence. Positive motivational counselling skills development [32] has been shown to improve provider-client relationships and subsequent retention and viral suppression [33]. Though primarily directed at lay counsellors, application of person-centered language and approaches across all facility staff cadres could positively affect client experience and thus long term and continuous engagement in care. Low-intensity models of care for established ART clients may reduce pressure on providers and allow them to spend more time with initiating and new patients [34].

Our results also identified financial constraints as a barrier to continuity of HIV care and treatment. Both the quantitative and qualitative results indicated that clients frequently found it difficult to obtain the cash needed to access health care at the prescribed intervals. Transport costs were most often noted, but other costs, such as child care, loss of income due to missing work for a day, and purchasing of food and drinks while waiting in queues, also present difficulties for clients who must make frequent clinic visits during the first 6 months on ART. While financial incentives have been shown to improve rates of retention and viral suppression in the first 6 months on ART [35], it is unclear whether these incentives result in sustained continuity of care and if an intervention like this could be sustainably scaled in a national program as large as South Africa's.

Many of the barriers that impede engagement in the early treatment period could potentially be addressed by existing service delivery options found within many countries' differentiated service delivery programs. Early period patients may benefit from fast-tracking, multi-month dispensing, and flexibility of location for medication pickups as much as, if not more than, established patients. Recognizing this potential, South Africa's most recent update to its national guidelines [36] introduces an initial viral load monitoring test at 3 months after initiation, rather than the usual six. This change effectively

brings forward eligibility for a differentiated model of service delivery forward to 4 months after initiation, the first routine clinic visit at which viral load test results are available. For clients who remain in care for the first four months, this is a promising improvement. As noted, however, most disengagement during the early treatment period occurs within the first 3 months.

Our study had a number of limitations. First, the sample size was modest, in terms of both number of study sites and number of participants, and generalizability to districts and provinces not included in the survey will require caution. Second, study clients were enrolled at any point up to 6 months after ART initiation, which may have created the possibility of recall bias for questions pertaining to their experiences and perceptions at ART initiation. As mentioned above, this enrollment strategy also potentially created a survival bias in our outcome results. We note, for example, that 86% of PREFER study participants had WHO stage recorded as Stage 1. This may reflect the need for clients to be well enough to participate in the study, resulting in exclusion of some of those with Stage 3 or 4 defining conditions. While we cannot exclude this possibility, we note that nearly a quarter (24%) of those enrolled presented with CD4 counts <200, the immunologic definition for advanced HIV disease, which is consistent with other estimates of AHD populations [37]. Third, we used participant self-report to determine prior ART use, which has previously been observed to be subject to much under-reporting; estimates suggest that anywhere from 20-50% of clients presenting for initiation of ART have previously used ART before [8]. To the extent to which this exposure misclassification occurred in the PREFER sample, our results will under-estimate the effect of prior ART use on risk of disengagement from care. Finally, we relied on routinely collected EMR data to ascertain participant ART retention outcomes at 6 months after ART initiation. This approach may also introduce bias due to outcome misclassification as our data is limited to visits observed at the initiating facility--we did not observe participants accessing care at other facilities. To the extent to which silent transfers such as these have occurred, our results will over-estimate rates of disengagement. Lastly, because FGDs were not stratified by initiating sub group (new, restart, <=6 mo) and we are unable to link individual responses within a FGD to their demographics, we could not examine differences within ART initiation sub-groups.

## Conclusions

Despite these limitations, this study identified a number of potentially modifiable factors that, if addressed, could improve early HIV treatment outcomes, in particular, improving clients' experiences at healthcare facilities and offering less expensive access to care, in terms of both time and transport costs. Accelerating eligibility for DSD models, so that even new ART clients can pick up medications at convenient locations and receive larger quantities of medications at a time, offers one way to address some of these issues. Strengthening the implementation of the 2023 revision to the National ART Guidelines [15], particularly with regard to CD4 count and viral load monitoring, could assist in identification and support of clients with advanced HIV disease and those eligible for earlier enrollment in differentiated service delivery options. While we did not identify a silver bullet for targeting clients at risk of disengagement, we began to describe subgroups who may benefit from more attention, such as previous disengagers. We hope that future research will further explore the characteristics of those who do, and do not, need intervention to achieve successful outcomes.

## Supporting information

**S1 Table. Participants' characteristics stratified by VL test result availability.**
(XLSX)

**S2 Table. Retention and VL suppression outcomes for participants who self-reported newly initiating treatment at study enrolment, stratified by their pattern of engagement (n = 296).**
(XLSX)

**S3 Table. Retention and viral load suppression outcomes for participants who self-reported re-engaging at study enrollment, stratified by their engagement pattern (n = 157).**
(XLSX)

**S4 Table. Retention and viral load suppression outcomes for participants who were on treatment (<6 months) at study enrollment, stratified by their engagement pattern (n = 596).**
(XLSX)

**S5 Table. Predictors of viral load suppression to <50 copies/mL by 6 months on ART.**
(XLSX)

## Acknowledgments

We would like to thank the clients and staff of the study sites for their cooperation in allowing us to conduct this study and the South Africa National Department of Health (NDOH) for approving this research.

## Author contributions

**Conceptualization:** Mhairi Maskew, Nancy Scott, Linda Sande, Sydney Rosen.

**Data curation:** Mhairi Maskew, Nyasha Mutanda, Vinolia Ntjikelane, Lufuno Malala, Musa Manganye.

**Formal analysis:** Nyasha Mutanda, Allison Morgan, Mariet Benade.

**Methodology:** Mhairi Maskew, Nancy Scott, Sydney Rosen.

**Supervision:** Mhairi Maskew, Sydney Rosen.

**Writing – original draft:** Mhairi Maskew, Nancy Scott, Allison Morgan, Sydney Rosen.

**Writing – review & editing:** Mhairi Maskew, Nyasha Mutanda, Mariet Benade, Vinolia Ntjikelane, Linda Sande, Lufuno Malala, Musa Manganye, Sydney Rosen.

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
