## [Decision Letter · Decision Letter 0]

21 Aug 2025

PGPH-D-25-01494

Drivers of disengagement from care during the first six months on antiretroviral therapy for HIV in South Africa: a mixed-methods study

Dear Dr. Maskew,

Thank you for submitting your manuscript to PLOS Global Public Health. After careful consideration, we feel that it has merit but does not fully meet PLOS Global Public Health’s publication criteria as it currently stands. Therefore, we invite you to submit a revised version of the manuscript that addresses the points raised during the review process.

We look forward to receiving your revised manuscript.

Kind regards,

Radhika Sundararajan

Academic Editor

Journal Requirements:

1. Please send a completed 'Competing Interests' statement, including any COIs declared by your co-authors. If you have no competing interests to declare, please state "The authors have declared that no competing interests exist". Otherwise please declare all competing interests beginning with the statement "I have read the journal's policy and the authors of this manuscript have the following competing interests:"

2. We noticed that you used “data not shown" in the manuscript. We do not allow these references, as the PLOS data access policy requires that all data be either published with the manuscript or made available in a publicly accessible database. Please amend the supplementary material to include the referenced data or remove the references.

3. Thank you for uploading your study's underlying data set. Unfortunately, the repository you have noted in your Data Availability statement does not qualify as an acceptable data repository according to PLOS's standards.

Additional Editor Comments (if provided):

Reviewers' comments:

Reviewer's Responses to Questions

**Comments to the Author**

1. Does this manuscript meet PLOS Global Public Health’s publication criteria?

Reviewer #1: Yes

Reviewer #2: Yes

2. Has the statistical analysis been performed appropriately and rigorously?

Reviewer #1: Yes

Reviewer #2: Yes

3. Have the authors made all data underlying the findings in their manuscript fully available (please refer to the Data Availability Statement at the start of the manuscript PDF file)?

Reviewer #1: Yes

Reviewer #2: Yes

4. Is the manuscript presented in an intelligible fashion and written in standard English?

Reviewer #1: Yes

Reviewer #2: Yes

Reviewer #1: Thank you for the opportunity to review this manuscript on the very important topic of early engagement on ART. The paper is very well written and mostly clear but I have some minor clarifying comments and edits.

Abstract

The preference data is mentioned in the methods but not the results. Could add to the “no differences” sentences for completeness.

Intro

Para starting line 42 – The way the groups are currently written it is not immediately clear if these are considered as mutually exclusive groups or more general risk groups that people may fall into. Someone may be a new or restarter with low CD4 for instance.

Also in this paragraph – perhaps add something to make clear why these groups are important. You have already spoken about how past interruption makes people more at risk for future interruption but what about the CD4 piece. Is starting ART at a low/high CD4 will be associated with higher risk of subsequent disengagement?

Methods

Study site – could you specify which provinces were included? With the large number of clinics I wondered if there were any differences by geography in any way. I see there were no differences in engagement by urban rural. Were there differences in the qualitative findings by urban/rural?

ART start groups - initiating, re-initiating, or on ART for ≤6 months: can you provide a little more information on the distinction between the on ART <=6 months groups and initiating group. Was there a minimum cutoff, perhaps after their first ART refill? Were any of these people very close to their ART start date (perhaps coming back for a blood result after 2 weeks or in clinic for something else shortly after). I understand this group is different potentially as you describe in the paper but it would help to describe the range of time they were on ART at enrolment.

Line 135 – counted tfo as continuously engaged but we know that many people get lost in transfer process. What about silent transfers – how common are they in your study settings?

Results

Line 249 – should this be among those who had a viral load (n=X)?

Similarly in line 259-261 it would be helpful to specify the number of people who had viral loads available. And in line 285 where you speak about predictors of VL<50.

Disengagement was defined as 28 days late for a scheduled visit. How accurate are the Tier.net data on scheduled visits and/or quantities of drug dispensed? I know the duration of follow-up is quite short but did results differ at all if you used a slightly longer window?

Did transfers differ at all by the predictors such as gender? I wonder if mobility, particularly unplanned, may be more common among in some groups, perhaps among men? Not to say that these people have not disengaged but there might be misclassification in assuming the transfers are retained and assuming the disengaged have not silently transferred. As raised earlier, it would be worth speaking to this in the discussion.

In the predictors section please add the RR and 95%CI for the age and cd4 associations that you mention in the text, even though they are not statistically significant, for consistency.

There is also some inconsistency in reporting the predictors of viral suppression, sometimes reporting % and sometimes RR and sometimes not reporting the results (CD4). This should be made consistent for ease of reading.

Did you compare the demographic characteristics of people with and without viral loads? Any differences in these groups? It would be good to see this in a supplemental table.

Line 358 typo: bring should be being

Were you able to draw out any differences/similarities in the qualitative findings based on the groups (new/restart/<=6m) and predictors in the quant analyses? It would be interesting to know if the qual findings differed at all by these subgroups or whether certain challenges appeared to come up more/less. In table 5, could you include the ART start group, age or age category, and gender of the participant?

Reviewer #2: To the authors:

Congratulations on this interesting and important study that focuses on patients’ experience during the early period after starting or re-starting ART. I enjoyed reading it. I have a few comments and questions that I hope will help the authors improve an already strong manuscript. I also noted a few minor language issues that need attention. Once addressed, I believe the manuscript will be appropriate for publication and has potential to add to the evidence on why individuals so frequently fail to remain on ART after beginning or re-starting treatment.

Specific comments:

Abstract

1. Page 2, lines 19-20: the Results section mentions alignment of findings with the socio-ecological model, but no mention of the model is made in the Methods. Could the authors describe (even briefly) the relevance of a socio-ecological model in analyses?

Introduction

2. Page 4: the paragraph starting with “In addition to the challenges associated with…” is not easy to follow because the 3 categories of “initiators” are distinctly different. Categories 1 and 3 are genuine initiators, but category 2 are described as individuals who are restarting following a lapse on treatment, so they are not truly “initiators.” It’s clear this manuscript will group these three categories for the purposes of the present analysis, but it would be a bit easier to read and follow if the description of the groups began with the true initiators (categories 1 and 3) and then explained that a third group (the re-starters) may be viewed as a type of treatment starter to better understand what people experience when they either start or re-start treatment. This flow seems more logical and would be easier to follow.

Methods

3. Page 6, line 84: how were the 18 healthcare facilities selected, and where were they located (e.g. nationwide or in specific areas)? Their client volumes were large, but can the authors describe other features that factored into the selection process so readers can better appreciate the study design and how similar the facilities were (e.g. public vs private, separate facility vs part of a larger hospital, in a rural vs urban setting)?

4. Page 7, lines 101-102: The sentence beginning with “We note that new guidelines were issued in 2023 and have revised the schedule for the early treatment period….” is a bit confusing – did the guidelines or the authors revise the schedule (reads like the latter but likely not accurate)?

5. Page 8, lines 127-128: what was the role of the second study research assistant?

6. Page 9, lines 164-167: “As the PREFER study aimed to understand client perceptions and experiences of HIV care throughout the first six months on ART, the study enrolled participants who had initiated ART up to 6 months prior to the study enrollment to ensure that the experiences and challenges of clients who had been accessing HIV care service delivery would be observed.” This describes an important feature of the design, and the purpose of the study. It would be helpful to readers if this point was made earlier in Methods or at the end of the Introduction to help clarify the purpose of the study and the range of participants being enrolled.

7. Pages 10-11, lines 184-194: There are many specific SEMs (e.g. McLeroy or the one developed by the CDC/NIH, to name two). Can the authors be more specific as to which one they employed for the analysis, why they chose it, and provide a citation?

Results

8. Page 12, lines 225-226: Table 2 provides data for the final linked sample and the qualitative FGD sample. Can the authors clarify what the “analytic sample” refers to in the sentence beginning with “The final linked sample….?

9. Page 13, Table 2: seeing the data on ART status at enrolment is a bit surprising given the attention to and language earlier in the Introduction on “initiation.” As it turns out, the study enrolled a far higher proportion of clients on already on treatment (57%) and re-engaging (15%) than actual ART initiators (28%). The vast majority of study participants thus had at least some experience with ART at the time of enrolment. The possibility that this would characterize the final study sample could be highlighted earlier in the description of the design and enrolment plan to alert readers that the study isn’t actually focused on first-time treatment initiators but rather on individuals either in an early stage of treatment or returning to care after some unknown prior period on ART (which might have been lengthy).

10. Page 18, lines 303-306. The sentence beginning with “Neither of these two….” is interpreting the results, which seems more appropriate for the Discussion section. Please consider moving it or elaborating on this point there.

11. Page 19, Table 5. There are numerous typos and grammar errors in the direct statements in the table. (Example: the second direct statement starting with “The medication intake….” where there are commas in places that should have periods. Another example: “My file was missing, I always have a new file and on subsequent visits the files will be all together and you just wonder where came from now.” This doesn’t make sense as written.) Given that the statements are the result of translations, it would be respectful to participants to present the statements here with correct grammar.

Minor comments – language and grammar

1. Page 4, line 40: The last sentence in the first paragraph of the Introduction ends with “….compared to those who had not interrupted.” This wording is a bit awkward. Consider “… compared to those who had not experienced a treatment interruption.”

2. Page 6: The sentence beginning with “During the study period, clients presenting for ART initiation or re-initiation or on ART for less than 6 months….” has a word or two missing before “on ART”.

3. Page 18, line 288. There is a missing period (“.”) at the end of the sentence before the one beginning with “Younger clients….”

**Do you want your identity to be public for this peer review?** For information about this choice, including consent withdrawal, please see our Privacy Policy

Reviewer #1: No

Reviewer #2: No

---

## [Editor Report · Decision Letter 1]

11 Nov 2025

PGPH-D-25-01494R1

Drivers of disengagement from care during the first six months on antiretroviral therapy for HIV in South Africa: a mixed-methods study

Dear Dr. Maskew,

Thank you for submitting your manuscript to PLOS Global Public Health. After careful consideration, we feel that it has merit but does not fully meet PLOS Global Public Health’s publication criteria as it currently stands. Therefore, we invite you to submit a revised version of the manuscript that addresses the points raised during the review process.

We look forward to receiving your revised manuscript.

Kind regards,

Radhika Sundararajan

Academic Editor

Journal Requirements:

Additional Editor Comments (if provided):

Please revise and resubmit with a response to the comments from the two reviewers. Your current cover letter does not include mention of the comments or how the manuscript was revised. The feedback from the initial review is re-copied below. If these revisions were made, please include a point-by-point response in your cover letter.

REVIEWER 1

Thank you for the opportunity to review this manuscript on the very important topic of early engagement on ART. The paper is very well written and mostly clear but I have some minor clarifying comments and edits.

Abstract

The preference data is mentioned in the methods but not the results. Could add to the “no differences” sentences for completeness.

Intro

Para starting line 42 – The way the groups are currently written it is not immediately clear if these are considered as mutually exclusive groups or more general risk groups that people may fall into. Someone may be a new or restarter with low CD4 for instance.

Also in this paragraph – perhaps add something to make clear why these groups are important. You have already spoken about how past interruption makes people more at risk for future interruption but what about the CD4 piece. Is starting ART at a low/high CD4 will be associated with higher risk of subsequent disengagement?

Methods

Study site – could you specify which provinces were included? With the large number of clinics I wondered if there were any differences by geography in any way. I see there were no differences in engagement by urban rural. Were there differences in the qualitative findings by urban/rural?

ART start groups - initiating, re-initiating, or on ART for ≤6 months: can you provide a little more information on the distinction between the on ART <=6 months groups and initiating group. Was there a minimum cutoff, perhaps after their first ART refill? Were any of these people very close to their ART start date (perhaps coming back for a blood result after 2 weeks or in clinic for something else shortly after). I understand this group is different potentially as you describe in the paper but it would help to describe the range of time they were on ART at enrolment.

Line 135 – counted tfo as continuously engaged but we know that many people get lost in transfer process. What about silent transfers – how common are they in your study settings?

Results

Line 249 – should this be among those who had a viral load (n=X)?

Similarly in line 259-261 it would be helpful to specify the number of people who had viral loads available. And in line 285 where you speak about predictors of VL<50.

Disengagement was defined as 28 days late for a scheduled visit. How accurate are the Tier.net data on scheduled visits and/or quantities of drug dispensed? I know the duration of follow-up is quite short but did results differ at all if you used a slightly longer window?

Did transfers differ at all by the predictors such as gender? I wonder if mobility, particularly unplanned, may be more common among in some groups, perhaps among men? Not to say that these people have not disengaged but there might be misclassification in assuming the transfers are retained and assuming the disengaged have not silently transferred. As raised earlier, it would be worth speaking to this in the discussion.

In the predictors section please add the RR and 95%CI for the age and cd4 associations that you mention in the text, even though they are not statistically significant, for consistency.

There is also some inconsistency in reporting the predictors of viral suppression, sometimes reporting % and sometimes RR and sometimes not reporting the results (CD4). This should be made consistent for ease of reading.

Did you compare the demographic characteristics of people with and without viral loads? Any differences in these groups? It would be good to see this in a supplemental table.

Line 358 typo: bring should be being

Were you able to draw out any differences/similarities in the qualitative findings based on the groups (new/restart/<=6m) and predictors in the quant analyses? It would be interesting to know if the qual findings differed at all by these subgroups or whether certain challenges appeared to come up more/less. In table 5, could you include the ART start group, age or age category, and gender of the participant?

6. PLOS authors have the option to publish the peer review history of their article (what does this mean?).

REVIEWER 2:

To the authors:

Congratulations on this interesting and important study that focuses on patients’ experience during the early period after starting or re-starting ART. I enjoyed reading it. I have a few comments and questions that I hope will help the authors improve an already strong manuscript. I also noted a few minor language issues that need attention. Once addressed, I believe the manuscript will be appropriate for publication and has potential to add to the evidence on why individuals so frequently fail to remain on ART after beginning or re-starting treatment.

Specific comments:

Abstract

1. Page 2, lines 19-20: the Results section mentions alignment of findings with the socio-ecological model, but no mention of the model is made in the Methods. Could the authors describe (even briefly) the relevance of a socio-ecological model in analyses?

Introduction

2. Page 4: the paragraph starting with “In addition to the challenges associated with…” is not easy to follow because the 3 categories of “initiators” are distinctly different. Categories 1 and 3 are genuine initiators, but category 2 are described as individuals who are restarting following a lapse on treatment, so they are not truly “initiators.” It’s clear this manuscript will group these three categories for the purposes of the present analysis, but it would be a bit easier to read and follow if the description of the groups began with the true initiators (categories 1 and 3) and then explained that a third group (the re-starters) may be viewed as a type of treatment starter to better understand what people experience when they either start or re-start treatment. This flow seems more logical and would be easier to follow.

Methods

3. Page 6, line 84: how were the 18 healthcare facilities selected, and where were they located (e.g. nationwide or in specific areas)? Their client volumes were large, but can the authors describe other features that factored into the selection process so readers can better appreciate the study design and how similar the facilities were (e.g. public vs private, separate facility vs part of a larger hospital, in a rural vs urban setting)?

4. Page 7, lines 101-102: The sentence beginning with “We note that new guidelines were issued in 2023 and have revised the schedule for the early treatment period….” is a bit confusing – did the guidelines or the authors revise the schedule (reads like the latter but likely not accurate)?

5. Page 8, lines 127-128: what was the role of the second study research assistant?

6. Page 9, lines 164-167: “As the PREFER study aimed to understand client perceptions and experiences of HIV care throughout the first six months on ART, the study enrolled participants who had initiated ART up to 6 months prior to the study enrollment to ensure that the experiences and challenges of clients who had been accessing HIV care service delivery would be observed.” This describes an important feature of the design, and the purpose of the study. It would be helpful to readers if this point was made earlier in Methods or at the end of the Introduction to help clarify the purpose of the study and the range of participants being enrolled.

7. Pages 10-11, lines 184-194: There are many specific SEMs (e.g. McLeroy or the one developed by the CDC/NIH, to name two). Can the authors be more specific as to which one they employed for the analysis, why they chose it, and provide a citation?

Results

8. Page 12, lines 225-226: Table 2 provides data for the final linked sample and the qualitative FGD sample. Can the authors clarify what the “analytic sample” refers to in the sentence beginning with “The final linked sample….?

9. Page 13, Table 2: seeing the data on ART status at enrolment is a bit surprising given the attention to and language earlier in the Introduction on “initiation.” As it turns out, the study enrolled a far higher proportion of clients on already on treatment (57%) and re-engaging (15%) than actual ART initiators (28%). The vast majority of study participants thus had at least some experience with ART at the time of enrolment. The possibility that this would characterize the final study sample could be highlighted earlier in the description of the design and enrolment plan to alert readers that the study isn’t actually focused on first-time treatment initiators but rather on individuals either in an early stage of treatment or returning to care after some unknown prior period on ART (which might have been lengthy).

10. Page 18, lines 303-306. The sentence beginning with “Neither of these two….” is interpreting the results, which seems more appropriate for the Discussion section. Please consider moving it or elaborating on this point there.

11. Page 19, Table 5. There are numerous typos and grammar errors in the direct statements in the table. (Example: the second direct statement starting with “The medication intake….” where there are commas in places that should have periods. Another example: “My file was missing, I always have a new file and on subsequent visits the files will be all together and you just wonder where came from now.” This doesn’t make sense as written.) Given that the statements are the result of translations, it would be respectful to participants to present the statements here with correct grammar.

Minor comments – language and grammar

1. Page 4, line 40: The last sentence in the first paragraph of the Introduction ends with “….compared to those who had not interrupted.” This wording is a bit awkward. Consider “… compared to those who had not experienced a treatment interruption.”

2. Page 6: The sentence beginning with “During the study period, clients presenting for ART initiation or re-initiation or on ART for less than 6 months….” has a word or two missing before “on ART”.

3. Page 18, line 288. There is a missing period (“.”) at the end of the sentence before the one beginning with “Younger clients….”
---

## [Editor Report · Decision Letter 2]

19 Nov 2025

Drivers of disengagement from care during the first six months on antiretroviral therapy for HIV in South Africa: a mixed-methods study

PGPH-D-25-01494R2

Dear Dr Maskew,

We are pleased to inform you that your manuscript 'Drivers of disengagement from care during the first six months on antiretroviral therapy for HIV in South Africa: a mixed-methods study' has been provisionally accepted for publication in PLOS Global Public Health.

Best regards,

Radhika Sundararajan

Academic Editor